# Not Every Dyspepsia Is Related to *Helicobacter pylori*—A Case of Esophageal Inlet Patch in a Female Teenager

**DOI:** 10.3390/children10020229

**Published:** 2023-01-28

**Authors:** Lorena Elena Meliț, Andreea Ligia Dincă, Reka Borka Balas, Simona Mocanu, Cristina Oana Mărginean

**Affiliations:** 1Department of Pediatrics I, George Emil Palade University of Medicine, Pharmacy, Science, and Technology of Târgu Mureș, Gheorghe Marinescu Street No 38, 540136 Târgu Mureș, Romania; 2Department of Pathology, County Emergency Hospital Târgu Mureș, Gheorghe Marinescu Street No 50, 540136 Târgu Mureș, Romania

**Keywords:** esophageal inlet patch, child, dyspepsia

## Abstract

*Helicobacter pylori* infection is one of the main causes of dyspepsia, but it is not the only cause. Esophageal inlet patches are areas of heterotopic gastric mucosa within the esophagus and are commonly located in the cervical part of the esophagus. We report the case of a 16-year-old female, previously known to display symptoms of anxiety, who was admitted to our clinic for dyspeptic symptoms lasting for approximately 1 month in spite of the treatment with proton pump inhibitors. The clinical exam revealed only abdominal tenderness in the epigastric area, while routine laboratory tests showed no abnormalities. The upper digestive endoscopy revealed a well-circumscribed salmon-pink-colored oval lesion of approximately 10 mm in the cervical esophagus, along with hyperemia of the gastric mucosa and biliary reflux. The histopathological exam established the diagnosis of esophageal inlet patch with heterotopic antral-type gastric mucosa and also revealed regenerative changes within the gastric mucosa. We continued to treat the patient with proton pump inhibitors, as well as ursodeoxycholic acid, with favorable evolution. Although rare or underdiagnosed, esophageal inlet patches should never be underestimated and all gastroenterologists should be aware of their presence when performing an upper digestive examination in a patient with dyspeptic symptoms.

## 1. Introduction

It is no longer a debatable topic that *Helicobacter pylori* (*H. pylori*), one of the most common bacterial infections worldwide, is commonly acquired during childhood, or that its long-term persistence might result in major complications due to chronic gastric inflammation; these complications might turn into several types of gastric cancer, such as gastric adenocarcinoma or gastric mucosa-associated lymphoid tissue lymphoma [1]. In fact, according to the World Health Organization, *H. pylori* associated gastric cancer is among the most frequent cancer-related causes of death, regardless of geographic area [2].

Although a lack of symptoms is commonly seen in children with *H. pylori* infections, dyspeptic symptoms such as nausea, vomiting, abdominal pain, or diarrhea might suggest the presence of this bacterium within the gastric mucosa [3]. Moreover, epigastric pain seems to be significantly associated with *H. pylori* positive gastritis [4,5]. Another challenge is related to the fact that, most often, children become asymptomatic shortly after the onset of these symptoms, if they even occur. Aside from the previously mentioned gastrointestinal symptoms, a recent review by our team found several extraintestinal disorders that might be related, though this is controversial. These disorders include iron deficiency anemia, purpura, growth retardation, cardiovascular diseases, metabolic syndrome, neurological disorders, dermatological conditions, ophthalmic diseases, or autoimmune pathologies [1].

Based on the above-mentioned controversies, the selection of the patients who would benefit from *H. pylori* testing is probably the most important step in the prevention of long-term associated complications. According to the Maastricht V/Florence Consensus Report, testing for *H. pylori* is recommended for patients with dyspepsia originating from high-prevalence areas; patients with peptic ulcers, particularly those with intake of aspirin or non-steroidal anti-inflammatory drugs, as well as individuals with a history of peptic ulcers; subjects with gastritis and long-term use of proton pump inhibitors; patients diagnosed with gastric cancer or who carry an increased risk of gastric cancer, along with those who were detected with localized early stage MALT lymphoma; and patients with different types of extraintestinal manifestations, such as iron deficiency anemia, vitamin B12 deficiency, and thrombocytopenic purpura without an identifiable cause [6]. It is important to mention that patients with extraintestinal manifestations might be considered a peculiar group for *H. pylori* testing, since it was recently emphasized that these manifestations, in spite of their extraintestinal pattern, seem to be closely related to the systemic subclinical inflammation triggered by an *H. pylori* infection; this relationship was detected even in pediatric patients [4,7]. Dyspepsia remains an important predictor of *H. pylori* infection.

Nevertheless, not everything in gastroenterology is about *H. pylori*, and not all patients with dyspepsia are infected with this bacterium. In fact, functional dyspepsia is among the most prevalent problems in pediatrics, and several gastrointestinal disorders aside from this infection were found to contribute to its occurrence. These include inflammatory conditions, gastrointestinal motility and sensory dysfunctions, gastrointestinal hormones, visceral hypersensitivity, and an altered brain–gut axis [8,9,10]. Functional dyspepsia is a syndrome defined by a group of gastrointestinal symptoms such as heartburn, epigastric or abdominal pain, early satiation, and postprandial fullness, with a prevalence reaching up to 57% in the general population [11]. Several risk factors were recently found to contribute to the occurrence of functional dyspepsia, such as living independently of parents, age, not eating breakfast, and the frequent consumption of cold or pickled foods [11].

Although all patients with dyspeptic symptoms must be screened for this infection, we should not forget about other causes of dyspepsia, especially those that are rare enough to be frequently overlooked. Esophageal inlet patches belong to this category of lesions, and they are defined as well-delineated oval, round, or geographically shaped areas of mucosa, which are salmon pink in color, presenting variable sizes. These lesions are commonly located within the cervical esophagus, and they might express different aspects, such as a smooth surface or a slightly elevated or depressed surface with heaped borders. On rare occasions, they may appear as polypoid or protrusive lesions. Small lesions might be easily overlooked, since they are often covered by esophageal squamous epithelium presenting no obvious changes in the overlying mucosa [12]. Although inlet patches are related to Barrett’s esophagus, they are usually overlooked, and their harmful impact might be underestimated, since they represent esophageal heterotopic gastric mucosa. These lesions can be congenital or acquired [13,14,15]. Reports on esophageal inlet patches remain scarce and they should definitely be studied in greater depth, since they might be involved in the etiology of different gastrointestinal symptoms.

The aim of this case report is to highlight a rare cause of dyspepsia that can be easily overlooked and to support the fact that *H. pylori* is not the only cause of dyspeptic symptoms.

Written informed consent was obtained from the patient’s mother prior to the publication of this case. 

## 2. Case Report

### 2.1. Presenting Concerns

We report the case of a 16-year-old female admitted to our clinic for nausea and epigastric pain she had experienced for approximately 1 month, for which proton pump inhibitors had been administered, but failed to improve her symptoms. The anamnesis revealed that the patient presented anxious elements.

### 2.2. Clinical Findings

The clinical exam at the time of admission revealed tenderness during palpation of the epigastric area.

### 2.3. Diagnostic Focus and Assessment

The routine laboratory tests performed on the day of admission revealed no abnormalities. The abdominal ultrasound found no pathological elements. We performed a rapid stool antigen test for *H. pylori* infection, but it was negative. Therefore, we performed an upper digestive endoscopy, which revealed hyperemic gastric mucosa and biliary reflux, but when we withdrew the endoscope, we also noticed a well-circumscribed salmon pink oval lesion of approximately 10 mm in the cervical esophagus (Figure 1). Several biopsies were obtained from both the esophageal lesion and the gastric mucosa. A histopathological exam showed regenerative changes within the gastric mucosa, while the esophageal lesion was suggestive of an inlet patch with the presence of antral heterotopic gastric mucosa. Therefore, we established a diagnosis of regenerative gastritis most likely caused by biliary reflux and an esophageal inlet patch.

### 2.4. Therapeutic Focus and Follow-Up

We recommended the continuation of the proton pump inhibitor treatment, as well as ursodeoxycholic acid, for the biliary reflux based on our previous clinical experience. The patient’s evolution was favorable after 1 month of treatment. Nevertheless, we will consider an endoscopic reevaluation of the inlet patch if the symptoms reoccur.

## 3. Discussion

The esophageal inlet patch was first described in 1805 in a postmortem examination of an esophagus, revealing aberrant gastric fundus-type epithelium in the upper esophagus [13,16]. In contrast, the inlet patch identified in our patient was detected with antral-type heterotopic gastric epithelium. Although the data on esophageal inlet patches so far result only from small studies and case reports, the prevalence of this lesion in the cervical esophagus varies between 0.18% and 14% [17]. However, autopsy reports reveal a significantly higher incidence of inlet patches, up to 70% [18], raising questions about whether these lesions are overlooked. Based on these discrepancies, these lesions might not be as rare as previously reported, since their detection depends on the thoroughness of the examiner when withdrawing the endoscope and requires a deep inspection of the proximal mucosa, or the use of advanced endoscopic techniques such as narrowband imaging [12]. Narrowband imaging has proven to be superior to other invasive diagnostic methods, since it provides a faster and more accurate diagnosis [19]. Moreover, studies on children have revealed this method’s advantage of accurately indicating the optimal areas for biopsy, demonstrating a strong correlation with the severity of the histopathological findings [20,21]. Thus, the detection rate of esophageal inlet patches could increase if the time spent withdrawing the endoscope and inspecting the proximal esophagus, which is often neglected, increases, or if the examiner has the opportunity to use narrowband imaging evaluation. In our case, although we did not use narrowband imaging, we noticed the lesion when we carefully withdrew the videoendoscope. Therefore, we re-emphasize the importance of thoroughness in endoscopic examination in order to avoid overlooking lesions located in unusual areas.

Although the pathogenesis of the cervical inlet patch remains unknown, a widely accepted hypothesis states that it has a congenital origin related to the incomplete transformation of columnar epithelium into squamous epithelium during embryonic development [22]. This hypothesis is supported by the immunohistochemically proven presence of glucagon-reactive cells, which are usually seen in the embryonic stage of gastric development [23]. Two less probable theories assume that these inlet patches might be the consequence of either 1) the rupture of an occluded proximal gland within the esophagus, contributing to the formation of retention cysts that burst and lead to the formation of heterotopic gastric mucosa [24], or 2) the metaplastic transformation of squamous epithelium into columnar epithelium as a consequence of acid exposure [25].

Several studies reported that esophageal inlet patches are more commonly found in males than in females [26,27]. Nonetheless, our patient was a female teenager presenting anxious elements. Taking into account that women more commonly present symptoms of anxiety, symptomatic inlet patches would be expected to be more common in females presenting with globus sensation, classified as type 2 following von Rahnen’s classification [15]. Similarly, other studies found a significant association between the length and size of inlet patches and globus sensation [28]. Moreover, Ciocalteu et al. noticed that symptomatic inlet patches are more frequent among females [12].

Another controversial topic is the relationship between *H. pylori* gastritis and symptomatic inlet patches. Some authors have hypothesized that gastroesophageal reflux is a mandatory condition for the colonization of inlet patches [29]. However, this is debatable, since other studies failed to identify a positive association between *H. pylori* and gastroesophageal reflux disease [30,31]. In our patient, we identified only biliary reflux, with no signs of gastroesophageal reflux disease. Conversely, other authors maintain that the symptoms associated with esophageal inlet patches, such as globus sensation and non-ulcer dyspepsia, might be related to *H. pylori* induced chronic inflammation within the gastric mucosa of the inlet patch [25]. These statements are further supported by Wüppenhorst et al., who described a case report of a patient whose symptoms improved after *H. pylori* eradication, indicating that the eradication of this infection might also result in important histopathological positive changes at the level of the colonized inlet patch mucosa [32]. Other reports suggest that the inlet patch mucosa has an up to 82% risk of being infected with *H. pylori* without any relationship between the type of inlet mucosa and the *H. pylori* colonization rate [33]. In contrast, Alagozlu et al. [25] noticed that *H. pylori* was more frequently identified in inlet patch fundic-type mucosa (81.2%). Our patient was identified as having antral-type gastric mucosa in the esophageal inlet patch, and we found no evidence of *H. pylori* infection either at this level or in the normal gastric mucosa. Moreover, the previously mentioned study stated that synchronous *H. pylori* positive gastritis was identified in all the patients in whom infected inlet patches were detected [25]. Based on these findings, the relationship between *H. pylori* infection and inlet patch-associated symptoms must still be clarified by further studies on a greater number of subjects.

Previous studies have emphasized a pathogenic link between inlet patches and Barrett’s esophagus due to their similarities regarding histopathological findings [34]; however, this was disputed by Feurle et al. [23], who concluded that they are histopathologically distinct, since inlet patches are characterized by embryonic cells while more primitive and multipotent cells characterize Barrett’s esophagus. We found no evidence of Barrett’s esophagus or gastroesophageal reflux in our patient. The pathogenic role of inlet patches goes further, since certain authors have emphasized these patches’ potential role in causing persistent dyspeptic symptoms or in favoring synchronous motility disorders [12,35]. Aside from the esophageal motor dysfunction that might be detected in patients with inlet patches, other factors were identified as contributing to motility dysfunction in these patients. These factors include decreased pressure in the lower esophageal sphincter; prolonged relaxation and lower amplitude of the peristaltic wave; prolonged exposure of the proximal and distal esophagus to gastric acid; and the presence of long-term biliary reflux within the distal esophagus [36]. Our patient also presented biliary reflux, which might have contributed to the presence of or worsened her dyspeptic symptoms. Less common symptoms include chest pain or shortness of breath, which might mimic cardiovascular disorders such as unstable angina pectoris, or swallowing difficulties [37].

Another controversial topic that requires further attention is the relationship between inlet patches and esophageal cancer, since no long-term studies have been performed on these patients in order to identify the precise role of these lesions in esophageal neoplasia [12]. Neoplastic progression of cervical inlet patches remains rare, with fewer than 50 cases of adenocarcinoma described in the literature to date [38]. As compared to Barrett’s esophagus, the lifetime incidence of neoplasia in patients with these lesions was proven to be lower: between 0 and 1.56% [25,26]. While the causative relationship between esophageal inlet patches and neoplastic transformation should be studied in more depth, it is clear that the presence of heterotopic gastric mucosa within the esophagus should be carefully searched for and addressed, especially in patients in whom the symptoms cannot be explained by other lesions in the upper gastrointestinal tract [12]. Several benign complications were also (though rarely) reported in patients with inlet patches, involving strictures, ulcerations, perforation, bleeding, and fistulas associated or unassociated with subcutaneous abscesses [18].

Unfortunately, no standard therapeutic approach is available for patients with esophageal inlet patches. Taking into account that the progression of these lesions to either benign or malignant complications is rare, asymptomatic patients in whom inlet patches are incidentally detected do not require any treatment, but they should be warned about possible risks at the time of diagnosis [39]. Moreover, the efficacy of proton pump inhibitors as an acid-suppressive therapy remains controversial [39]. Endoscopy-based techniques proved to be the most useful in patients with symptomatic inlet patches. According to multiple studies and case reports, argon plasma coagulation resolved symptoms such as globus sensation in up to 82% of patients, resulting in complete endoscopic healing in 90% of these cases [40,41,42]. Endoscopic mucosal resection or submucosal dissection were mainly used in patients experiencing a malignant transformation of an inlet patch, but the data regarding their long-term efficacy remain scarce [43,44]. A novel therapeutic approach involves radiofrequency ablation, which proved a complete endoscopic and histological resolution of the lesions in 80% of cases, while 20% of the patients showed over 90% resolution [45]. The same study indicated a considerable improvement in patients’ symptoms (globus sensation, cough, and sore throat) as a result of radiofrequency ablation [45]. The dyspeptic symptoms experienced by our patient was most likely related to her biliary reflux and not the inlet patch, since they resolved after treatment. Thus, we might consider that the inlet patch was an incidental finding in our case, and we do not consider that the patient required any therapeutic approach for this lesion.

## 4. Conclusions

Esophageal inlet patches, especially those located in the cervical esophagus, remain frequently undiagnosed due to the lack of thorough inspection of the esophagus during an upper digestive endoscopy. Although these lesions usually appear in the presence of heterotopic gastric mucosa within the esophagus, their relationship with the patient’s dyspeptic symptoms remains controversial and their colonization by *H. pylori* should be ruled out in all cases in order to prevent further long-term life-threatening complications such as malignant transformation. A more in-depth analysis of these lesions in larger studies, along with long-term follow-up, is required in order to identify their precise pathogenic role, but regardless, inlet patches should never be overlooked.

## Figures and Tables

**Figure 1 children-10-00229-f001:**
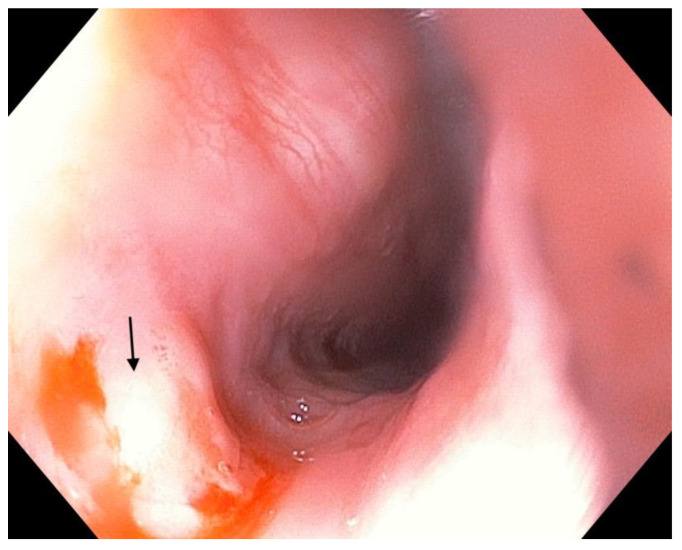
Macroscopic aspect of the inlet patch.

## Data Availability

Not applicable.

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
