# Peer review of "Not Every Dyspepsia Is Related to Helicobacter pylori—A Case of Esophageal Inlet Patch in a Female Teenager"

_children, 2023, doi:10.3390/children10020229_

Round 1

Reviewer 1 Report

Melit et al investigated the relationship between esophageal inlet lesions and resulting dyspepsia through a case study and reviewed relevant literature. The study is thorough and the manuscript is well-written. 

Author Response

January the 23rd, 2023

To Editor of Children,

Dear Editor,

Please find attached a revised version of the manuscript entitled: ‘Not every dyspepsia is related to Helicobacter pylori – a rare case of esophageal inlet patch in a teenager Lorena Elena MeliÈ›, Andreea Ligia Dincă, Reka Borka Balas, Simona Mocanu and Cristina Oana Mărginean

Manuscript ID: 2109314 

Firstly, we thank very much the reviewers for their valuable comments and suggestions in order to improve our paper.

Following the reviewers’ concerns and observations, we made some modifications to the initial version of our manuscript, which we described in great detail, according to their recommendations, as it follows:

COMMENTS TO AUTHOR:

Reviewer 1

Comment 1:

Melit et al investigated the relationship between esophageal inlet lesions and resulting dyspepsia through a case study and reviewed relevant literature. The study is thorough and the manuscript is well-written.

Answer 1: Thank you for your positive comments and your valuable time spent on assessing our manuscript.

We corrected the manuscript after being revised by a native English speaker.

We thank you for all your valuable comments and the time you spent on assessing our manuscript.

Respectfully,

Professor Cristina Oana Marginean, MD, PhD

Lecturer Lorena Elena Melit, MD, PhD

Reviewer 2 Report

The manuscript "Not every dyspepsia is Helicobacter pylori-a case of esophageal inlet patch and a review of the literature" explores a very interesting and didactic case regarding differential diagnosis of dyspepsia. I suggest some comments that could improve de quality of the manuscript:

1.       Abstract:

·       Line 23: I suggest to change the expression “a relative of Barrett´s esophagus”. Relationship between Barrett´s esophagus and esophageal inlet patch is controversial at the moment and we can not confirm that both entities share etiopathogenesis or risk of neoplastic degeneration. Therefore, this expression could be confusing.

2.       Introduction:

·       Introduction is mainly focused on H. pylori infection. It should address the differential diagnosis of dyspepsia from a broader perspective.

·       Paragraphs are too long and it is difficult to read them

·       Please, I encourage the authors to edit lines 46-56

·       Please, change the expresion “MALToma” for MALT lymphoma

3.       Case reports:

·       Line 109: Please, correct “Figure no 1”

·       Figure 1:  Could you provide a higher quality image? If not, I recommend to the authors to point the lesion with an arrow. Was the image taken after having biopsied the lesion (presence of blood)? If the lesion was bleeding spontaneously please specify it in the description.

·       On what evidence was the initiation of treatment with ursodeoxycholic acid supported? Literature should be cited. If not, please specify that it was based on clinical experience/empirical treatment.

4.       Discussions:

·       Lines 125 and 128: do not repeat the same connectors

·       Are the patient´s dyspeptic symptoms likely to be attributed to the esophageal inlet patch? Could esophageal inlet patch be an incidental finding? This reflexion should be included in discussion section

5.       Line 116: Why informed consent was not applicable? Is there written consent to publish the endoscopic image?

Thanks for the opportunity of review this interesting manuscript.

Author Response

January the 23rd, 2023

To Editor of Children,

Dear Editor,

Please find attached a revised version of the manuscript entitled: ‘Not every dyspepsia is related to Helicobacter pylori – a rare case of esophageal inlet patch in a teenager Lorena Elena MeliÈ›, Andreea Ligia Dincă, Reka Borka Balas, Simona Mocanu and Cristina Oana Mărginean

Manuscript ID: 2109314 

Firstly, we thank very much the reviewers for their valuable comments and suggestions in order to improve our paper.

Following the reviewers’ concerns and observations, we made some modifications to the initial version of our manuscript, which we described in great detail, according to their recommendations, as it follows:

Reviewer 2

Comment 1

The manuscript "Not every dyspepsia is Helicobacter pylori-a case of esophageal inlet patch and a review of the literature" explores a very interesting and didactic case regarding differential diagnosis of dyspepsia. I suggest some comments that could improve de quality of the manuscript:

  1. Abstract:

Line 23: I suggest to change the expression “a relative of Barrett´s esophagus”. Relationship between Barrett´s esophagus and esophageal inlet patch is controversial at the moment and we can not confirm that both entities share etiopathogenesis or risk of neoplastic degeneration. Therefore, this expression could be confusing.

Answer 1

Thank you for your suggestion. We rephrased as following: ‘Although rare or underdiagnosed, esophageal inlet patches should never be underestimated and each gastroenterologist should be aware of their presence when performing an upper digestive examination in a patient with dyspeptic symptoms.’

Comment 2

Introduction:

  • Introduction is mainly focused on H. pylori infection. It should address the differential diagnosis of dyspepsia from a broader perspective.
  • Paragraphs are too long and it is difficult to read them
  • Please, I encourage the authors to edit lines 46-56
  • Please, change the expresion “MALToma” for MALT lymphoma

Answer 2

Thank you for your suggestions.

  • We focused mainly on H. pylori-related dyspepsia since our case was meant to underline that not all cases with dyspepsia are due to this infection. Nevertheless, we introduced several details regarding other causes of dyspepsia: ‘In fact, functional dyspepsia is among the most prevalent problems in pediatrics and several gastrointestinal disorders aside from this infection were found to contribute to its occurrence such as inflammatory conditions, gastrointestinal motility and sensory dysfunctions, gastrointestinal hormones, visceral hypersensitivity, as well as altered brain-gut axis (Keely S, Walker MM, Marks E, Talley NJ. Immune dysregulation in the functional gastrointestinal disorders. Eur J Clin Invest. 2015;45:1350–9; Madisch A, Andresen V, Enck P, Labenz J, Frieling T, Schemann M. The diagnosis and treatment of functional dyspepsia. Dtsch Arztebl Int. 2018;115:222–32; Cui J, Wang J, Wang Y, Zhang C, Hu G, Wang Z. External treatment of traditional Chinese medicine for functional dyspepsia in children: Protocol for a systematic review and network meta-analysis. Medicine 2022;101:43(e31597).’
  • We split the paragraphs for clarity as you recommended
  • We rephrases lines 45-46 as you suggested: ‘Another challenge is related to the fact that most often children become asymptomatic shortly after the onset of these symptoms if they even occur.’
  • We changed the expression according to your recommendations

Comment 3

Case reports:

  • Line 109: Please, correct “Figure no 1”
  • Figure 1: Could you provide a higher quality image? If not, I recommend to the authors to point the lesion with an arrow. Was the image taken after having biopsied the lesion (presence of blood)? If the lesion was bleeding spontaneously please specify it in the description.
  • On what evidence was the initiation of treatment with ursodeoxycholic acid supported? Literature should be cited. If not, please specify that it was based on clinical experience/empirical treatment.

Answer 3

Thank you for your remarks

  • We corrected ‘Figure 1’
  • The image was taken with the videoendoscope and its quality cannot be modified, but we pointed with an arrow the lesion
  • The treatment with ursodeoxycholic acid was indeed based on our clinical experience since we previously used this drug in children with biliary reflux and they experienced a favorable evolution. We also mentioned this fact in the text

Comment 4

Discussions:

  • Lines 125 and 128: do not repeat the same connectors
  • Are the patient´s dyspeptic symptoms likely to be attributed to the esophageal inlet patch? Could esophageal inlet patch be an incidental finding? This reflexion should be included in discussion section

Answer 4

Thank you again for your suggestions.

  • We replaced the repeated word ‘Contrariwise’ with ‘Nevertheless’
  • We included this reflexion in the discussions section: ‘The dyspeptic symptoms experienced by our patients were most likely related to the biliary reflux and not the inlet patch since they resolved after treatment. Thus, we might consider that the inlet path was an incidental finding.

Comment 5

  1. Line 116: Why informed consent was not applicable? Is there written consent to publish the endoscopic image?

Answer 5

We apologize for our typographical mistake. The informed consent was obtained from the patient’s mother prior to the publication of this case and we corrected in the revised form of our manuscript this statement. In fact, the policy of our hospital implemented a consent form in each clinical chart and it is usually signed by the child’s care-givers at the time of admission specifying that they agree with the publication of their child’s case for scientific purpose.

Comment 6

Thanks for the opportunity of review this interesting manuscript.

Answer 7

We deeply appreciate all your valuable suggestions and recommendations and we hope to have fulfilled your requests.

We corrected the manuscript after being revised by a native English speaker.

We thank you for all your valuable comments and the time you spent on assessing our manuscript.

Respectfully,

Professor Cristina Oana Marginean, MD, PhD

Lecturer Lorena Elena Melit, MD, PhD

Reviewer 3 Report

Dear Authors,

Thank you for the manuscript, please consider the following points 

1- I don't see a real discussion for the previous researchs , as you mentioned this is a literature review it is just a case report with additional points ??!

2- Why you don't want to consider a re-endoscopy till the symptoms back? do you think that if the symptoms don't reoccur that means the patient was cured??

3- Why is the presence of HP in this inlent patch in the oesophagus isn't from a biopsy infection from the stomach when you collect the biopsies? 

4- Please we need more specifications regarding the title, the title does not describe clearly the content

5- The introduction is very wide, you tried to suggest a historical linkage between HP, dyspepsia, esophagus inlent  patch, I think you need to be more specified

6- English changes is required 

Author Response

January the 23rd, 2023

To Editor of Children,

Dear Editor,

Please find attached a revised version of the manuscript entitled: ‘Not every dyspepsia is related to Helicobacter pylori – a rare case of esophageal inlet patch in a teenager Lorena Elena MeliÈ›, Andreea Ligia Dincă, Reka Borka Balas, Simona Mocanu and Cristina Oana Mărginean

Manuscript ID: 2109314 

Firstly, we thank very much the reviewers for their valuable comments and suggestions in order to improve our paper.

Following the reviewers’ concerns and observations, we made some modifications to the initial version of our manuscript, which we described in great detail, according to their recommendations, as it follows:

Reviewer 3

Comment 1

Thank you for the manuscript, please consider the following points 

1- I don't see a real discussion for the previous researches, as you mentioned this is a literature review it is just a case report with additional points ??!

Answer 1

According to your remark, we deleted a literature review from the title.

Comment 2

2- Why you don't want to consider a re-endoscopy till the symptoms back? do you think that if the symptoms don't reoccur that means the patient was cured??

Answer 2

According to the data available in the literature, most often inlet patches are asymptomatic and they do not require re-biopsy only in the case when children become symptomatic. Based on the fact that these lesions only in extremely rare cases experience malignant transformation and this occurs only in those with a certain trigger (gastroesophageal reflux, H. pylori infection of the inlet patch, etc.), that results in certain symptoms, re-endoscopy should be performed only in selected cases. Moreover, none of the reports of the literature recommends re-biopsy or re-endoscopy in asymptomatic patients.

Comment 3

3- Why is the presence of HP in this inlent patch in the oesophagus isn't from a biopsy infection from the stomach when you collect the biopsies? 

Answer 3

We apologize for not stating more clear that we took biopsies from both inlet patch and gastric mucosa and none of them proved to be infected with H. pylori according to the histopathology exam.

Comment 4

4- Please we need more specifications regarding the title, the title does not describe clearly the content

Answer 4

The title is meant to be mysterious in order to attract the readers to see why not all cases of dyspepsia are related to H. pylori infection and what is an inlet patch, and we consider it would be extremely valuable for this case since our intention was to underline that not dyspepsia is not always related to H. pylori infection. Nevertheless, we rearranged the title in order to be more clear: ‘Not every dyspepsia is related to Helicobacter pylori – a rare case of esophageal inlet patch in a teenager’.

Comment 5

5- The introduction is very wide, you tried to suggest a historical linkage between HP, dyspepsia, esophagus inlent  patch, I think you need to be more specified

Answer 5

According to your recommendations, we deleted the unnecessary details from the introduction section. Nevertheless, we considered it is important to keep the structure since the aim of this review was to underline that H. pylori is not always responsible for the patient’s dyspepsia and we must also consider more rare pathologies such as inlet patch. Nevertheless, in order to differentiate the causes of dyspepsia, we must know all the details regarding H. pylori clinical picture.

Comment 6

6- English changes is required 

Answer 5

We corrected the manuscript after being revised by a native English speaker.

We thank you for all your valuable comments and the time you spent on assessing our manuscript.

Respectfully,

Professor Cristina Oana Marginean, MD, PhD

Lecturer Lorena Elena Melit, MD, PhD

Round 2

Reviewer 3 Report

Dear Authors,

Thank you for considering my comments, well done 

Best regards